# Ploidy Status, Nuclear DNA Content and Start Codon Targeted (SCoT) Genetic Homogeneity Assessment in *Digitalis purpurea* L., Regenerated In Vitro

**DOI:** 10.3390/genes13122335

**Published:** 2022-12-11

**Authors:** Yashika Bansal, A. Mujib, Zahid H. Siddiqui, Jyoti Mamgain, Rukaya Syeed, Bushra Ejaz

**Affiliations:** 1Cellular Differentiation and Molecular Genetics Section, Department of Botany, Jamia Hamdard, New Delhi 110062, India; 2Genomic and Biotechnology Unit, Department of Biology, Faculty of Science, University of Tabuk, Tabuk 71491, Saudi Arabia

**Keywords:** indirect somatic embryogenesis, shoot organogenesis, nuclear DNA content, genetic homogeneity, SEM, flow cytometry, SCoT marker

## Abstract

*Digitalis purpurea* L. is a therapeutically important plant that synthesizes important cardiotonics such as digitoxin and digoxin. The present work reports a detailed and efficient propagation protocol for *D. purpurea* by optimizing various PGR concentrations in Murashige and Skoog (MS) medium. The genetic homogeneity of in vitro regenerants was assessed by the flow cytometric method (FCM) and Start Codon Targeted (SCoT) marker technique. Firstly, the seeds inoculated in full MS medium added with 0.5 mg/L GA_3_ produced seedlings. Different parts such as hypocotyl, nodes, leaves and apical shoots were used as explants. The compact calli were obtained on BAP alone or in combinations with 2, 4-D/NAA. The hypocotyl-derived callus induced somatic embryos which proliferated and germinated best in 0.75 mg/L BAP-fortified MS medium. Scanning electron microscopic (SEM) images confirmed the presence of various developmental stages of somatic embryos. Shoot regeneration was obtained in which BAP at 1.0 mg/L and 2.0 mg/L BAP + 0.5 mg/L 2,4-D proved to be the best treatments of PGRs in inducing direct and indirect shoot buds. The regenerated shoots showed the highest rooting percentage (87.5%) with 24.7 ± 1.9 numbers of roots/shoot in 1.0 mg/L IBA augmented medium. The rooted plantlets were acclimatized in a greenhouse at a survival rate of 85–90%. The genome size and the 2C nuclear DNA content of field-grown, somatic embryo-regenerated and organogenic-derived plants were estimated and noted to be 3.1, 3.2 and 3.0 picogram (pg), respectively; there is no alteration in ploidy status and the DNA content, validating genetic uniformity. Six SCoT primers unveiled 94.3%–95.13% monomorphic bands across all the plant samples analyzed, further indicating genetic stability among in vitro clones and mother plants. This study describes for the first time successful induction of somatic embryos from hypocotyl callus; and flow cytometry and SCoT marker confirmed the genetic homogeneity of regenerated plants.

## 1. Introduction

*Digitalis purpurea* L., a member of the Plantaginaceae family, is a biennial/perennial herbaceous plant [1]. It is popularly known as foxglove (lady’s glove). *D. purpurea* is an indigenous species of Europe which has now been widely cultivated in various parts of the world, such as China, New Zealand, USA and Canada. It has conspicuous bell-shaped flowers of different colors ranging from purple to yellow, pink, gray or white and hence, it is being grown as an ornamental plant across the world [2]. Apart from ornamental values, the plant also holds great medicinal importance in the pharmaceutical sector due to the presence of cardiac glycosides. The members of this cardenolide group include digitoxin, digoxin, digitoxigenin, digoxigenin, medigoxin, strophanthins and lanthosides [3]. Digitoxin, digoxin and lanthosides are the major bioactive compounds found in *D. purpurea* [4]. These cardenolides have been extensively utilized in treating congestive heart failure, myocardial infarction, oedema, coronary artery disease, hypertrophy and hypertension [5]. In recent times, digitoxin and digoxin have proved to be effective as anti-cancer and anti-viral agents [6].

*Digitalis* is the natural source of glycosides, and it is quite difficult to synthesize under laboratory conditions owing to structural intricacy. The unregulated utilization of this plant by the pharmaceutical industry has been a major cause of the decline of the plant population in its native environment [7]. The delayed seed-set formation, low seed germination and viability limit the growth in the wild [8,9]. Under such circumstances, plant tissue culture can serve as a better alternative for rapid multiplication of microbe-free plants in a short period of time, which is otherwise not possible to achieve under natural conditions [10]. The in vitro culture technique offers several advantages over conventional breeding approaches such as fast propagation, preservation of germplasm, production of polyploids, genetic transformation and crop advancement [11,12]. In addition, it provides continual and enhanced yields of medically important bioactives found in plants [13].

Somatic embryogenesis and organogenesis are the two intriguing biotechnological approaches to produce in vitro regenerated plantlets [14]. However, due to prolonged exposure to various Plant Growth Regulators (PGRs), continuous passaging and genotypic make-up, the regenerated plants often accumulate genetic and epigenetic alterations, commonly termed as somaclonal variations [15,16]. Therefore, the genetic uniformity evaluation of plants raised in vitro is crucial to ensure commercial uses on a global level [17,18]. In this regard, flow cytometric method proves to be a handy technique for measuring nuclear 2C DNA content of plant cells [19], making it simple, fast and easy to determine the ploidy of micropropagated plants [20]. Recent investigations have been done to examine the ploidy levels in various plants, such as *Allium sativum* [21], different species of *Zephyranthes* [22] and *Catharanthus roseus* [23]. Additionally, the genetic homogeneity of tissue culture plants can also be carried out by using PCR-based DNA markers such as Random Amplified Polymorphic DNA (RAPD), Start Codon Targeted (SCoT), Inter Simple Sequence Repeat (ISSR) and Amplified Fragment Length Polymorphism (AFLP) [24]. The SCoT marker technique ascertains polymorphism in the genome of micropropagated plants, thereby screening out possible somaclonal variations with high efficacy [25], as reported successfully in different plants [26,27].

In this current study, a sustainable in vitro plant propagation protocol via somatic embryogenesis and shootlet organogenesis in *D. purpurea* has been devised. To date, there are no reports of somatic embryo formation in *D. purpurea;* thus, for the first time, this article describes an efficient embryogenesis method from hypocotyl callus. Different morphological developments of somatic embryos have also been described using scanning electron microscopy (SEM). Moreover, this is the first ever report of assessing genetic homogeneity of in vitro-derived plantlets in *D. purpurea* with respect to the mother plant using flow cytometry and SCoT molecular marker approaches.

## 2. Materials and Methods

### 2.1. Collection of Seed Materials and Explant Preparation

The seeds of *Digitalis purpurea* L. were collected from the Herbal Research and Development Institute (HRDI), Mandal (coordinates: 30°27′13.3″ N 79°16′17.8″ E), Chamoli district of Uttarakhand, India. Initially, the collected seeds were immersed in a 20% teepol solution for 10 min, then washed under running tap water for several minutes. Afterwards, the seeds were surface-sterilised with 70% ethanol and 0.1% HgCl_2_ for 2 min each, and then thoroughly rinsed with sterilised double-distilled water three times to eliminate remnants of the sterilising agents.

### 2.2. Seed Germination and Culture Conditions

For the effectual germination, the disinfected seeds were then aseptically placed onto full- and half-strength Murashige and Skoog (MS) basal medium [28] comprised of 3% (*w*/*v*) sucrose and 0.8% (*w*/*v*) agar. The pH calibration of the medium was done at 5.7 using 1N HCl and/or 1N NaOH preparatory to autoclaving at 121 °C for 15 min. All culture tubes were maintained at 25 ± 2 °C under white fluorescent light (irradiance at 50 μmol/m^2^/s^−1^) for a 16 h photoperiod at a relative humidity of 60%. The frequency of seed germination was noted after a three-week period. Node, leaf, hypocotyl and apical shoot segments from 21-day-old germinated seedlings were taken as explants for further experimentation.

### 2.3. Callus Induction and Proliferation

Different explants, such as node, leaf and hypocotyl, were inoculated on MS medium enriched with 0.5–2.0 mg/L BAP (6-benzylaminopurine) alone/in combination with varied 2,4-Dconcentrations (2,4-dichlorophenoxyacetic acid)/NAA (α-naphthalene acetic acid) for callus induction. The cultures were maintained by sub culturing the callus in the same PGR added MS medium after 3–4 weeks The callus induction frequencies of different explants used were recorded three weeks after the culture period. The texture of the callus was also noted for each explant used in the experiments.
Callus induction frequency (%)=Number of explants showing callusing÷Total number of explants inoculated×100

### 2.4. Indirect Somatic Embryogenesis

Within 4–5 weeks of continuous sub-culturing of hypocotyl-derived callus on MS, the non-embryogenic calli were transformed into greenish embryogenic calli, distinguished by the presence of different embryo stages, such as globular, heart- or torpedo-shaped. The somatic embryogenesis percentage and the obtained somatic embryos per hypocotyl-derived callus were noted after five weeks of culture. The germination of somatic embryos was accomplished either on full MS fortified with low levels of BAP or without any PGR, and the germination frequency was recorded after five weeks of culture.

### 2.5. Scanning Electron Microscopy (SEM)

The origin and different developmental stages of somatic embryos were analyzed using scanning electron microscopy. For this purpose, the embryonic calli were prefixed with 2.5% (*v*/*v*) glutaraldehyde and 0.1 M of phosphate buffer (pH 6.8) and incubated for 24 h at 4 °C. The samples were then rinsed with the same buffer and postfixed with 0.1% fresh osmium tetroxide for 2 h. Thereafter, the calli were subject to graded ethanol dehydration series [25% (*v*/*v*), 50% (*v*/*v*), 75% (*v*/*v*), 90% (*v*/*v*)] for 15 min each and 100% ethanol twice for 12–15 min. Later, the dried tissues underwent sputter coating with a gold-palladium alloy and finally viewed under a LEO 435 VP scanning electron microscope (Zeiss, Oberkochen, Germany) operated at 20 kV.

### 2.6. Shoot Organogenesis via Direct and Indirect Method

Apical buds and cotyledonary nodal segments from 21-day-old in vitro raised seedlings were taken as explants for direct shoot regeneration. The explants were inoculated onto shoot multiplication MS medium augmented with varied concentrations of BAP (0.5–2.0 mg/L) or kinetin (Kn) (0.5–2.0 mg/L). The passaging of the cultures in the same medium was repeatedly done every three weeks. The percentage of shoot proliferation (total number of shoots emerged/total explants × 100) and the average regenerated shoot length per treatment were counted after five weeks.

To induce indirect shoots, proliferative calli obtained from leaf and hypocotyl explants were transferred to MS added with different concentrations and combinations of BAP (0.5–2.0 mg/L) and 2,4-D (0.5–1.0 mg/L). The shoot induction rate (%) and the mean number of shoots/callus obtained were counted after 5 weeks.

### 2.7. Root Initiation and Acclimatization

In vitro regenerated shoots (3–4 cm long), derived from somatic embryos and organogenic callus, were excised and transferred onto root-inducing MS medium incorporated with various concentration ranges of indole-3-acetic acid (IAA)/indole-3-butyric acid (IBA)/NAA. Data pertaining to the effects of different auxin treatments were noted down as root induction frequency (%) and the average number of roots generated by each shoot after three weeks of culture. The rooted plantlets were subsequently washed with autoclaved double-distilled water to clear away the adhering culture medium, and shifted to plastic pots carrying an autoclaved mixture of sand, soil and soilrite (1:1:1). Initially, these potted plants were wrapped with transparent polybags and maintained in a culture room (temperature 25 ± 2 °C; humidity 70 ± 10%; light 60 μmolm^−2^s^−1^) for two weeks. Afterwards, the plants were allowed to grow under greenhouse conditions with an optimum temperature of 27 ± 2 °C, 60–70% relative humidity and a 10–12 h photoperiod.

### 2.8. Flow Cytometry

Flow cytometric analyses were conducted to determine the genetic stability and the genome size of in vitro regenerated plantlets according to the method described by Galbraith [29]. Fresh young leaves were randomly selected from field-grown *Digitalis* (control) and plants regenerated via somatic embryogenesis and indirect organogenesis for ploidy level analysis. *Digitalis lanata* Ehrh. was used as a reference standard with known 2C nuclear DNA content of 3.0 pg [30]. Approximately 100 mg of each leaf sample was finely chopped using a fresh surgical blade in a pre-chilled Petri-dish containing 1.5 mL ice-cold Galbraith’s buffer (45 mM MgCl_2_, 30 mM sodium citrate, 20 mM MOPS and 0.1% Triton-X) for nuclei extraction. Afterwards, the suspensions were filtered using a double-layered nylon mesh of 50 μm pore size to remove larger cellular debris and were stained with 50 μg/mL of PI RNase (Propidium iodide RNase) (Sigma-Aldrich, St. Louis, MO, USA) for 10 min. The samples were then put in dark conditions at 4 °C for about 1 h. Finally, the incubated samples were analyzed on a BD FACS (Calibur) flow cytometer (BD Biosciences, Franklin Lakes, NJ, USA). The 2C DNA content of regenerated plants of *D. purpurea* was estimated by using the formula [31]:
2C DNA content of sample (pg) = 2C DNA content of standard (pg) × mean position of G_0_/G_1_ peak of sample
mean position of G_0_/G_1_ peak of standard

### 2.9. SCoT Marker Analysis

The total genomic DNA extraction was done from 500 mg of each leaf sample of twelve randomly selected in vitro regenerants (six derived from somatic embryos and six from indirectly induced shoots) along with the mother plant by following the modified cetyltrimethyl ammonium bromide (CTAB) methodology [32]. The quantitative and qualitative parameters of isolated DNA were assessed by agarose gel electrophoresis (0.8%).

For genetic homogeneity studies, a total of 12 SCoT primers were examined, out of which six produced desirable and reproducible amplified bands, which were finally used for further amplification reactions). The SCoT-PCR reaction mixture of 15 μL contained 50–60 ng of genomic DNA, 10× Taq polymerase buffer, 2.5 mM MgCl_2_, 10 mM dNTPs, forward and reverse primer (18 nucleotides long), 5 units/μL of Taq DNA Polymerase (Sigma-Aldrich) and deionised water. The PCR technique was operated in a thermal cycler (GeneAmp PCR 9700) with an initial denaturation at 94 °C for 5 min, followed by 35 cycles of 30 s denaturation at 94 °C, 30 s annealing of primers at 50 °C and 1 min extension at 72 °C with the final extension for 5 min at 72 °C. The amplified products were then cooled down at 4 °C (holding temperature) and resolved through agarose gel electrophoresis (1.5%) by using 1X TBE (Tris Borate EDTA) buffer. All the amplification reactions with SCoT primers were replicated thrice to confirm the reproducibility. The gel images were captured using a gel documentation system (Azure Biosystem, Dublin, CA, USA). The sizes of the PCR amplicons were determined by 1 kb DNA ladder (Gene DireX, Inc., Taoyuan, Taiwan).

The distinguishable and reproducible bands produced by SCoT primers were scored manually for their presence and absence. The genetic similarity (GS) values between mother plant and in vitro-derived plantlets were computed based on the Jaccard’s similarity coefficient. The obtained similarity coefficients were then utilized to construct a dendrogram through NTSYSpc software (version 2.02, Rohlf, New York, USA) [33] using the UPGMA (Unweighted Pair Group Method of Arithmetic Averages) method [34].

### 2.10. Statistical Analysis

The in vitro experiments were performed in a completely randomized design (CRD). The data influencing the role of PGRs on explants in inducing callus, somatic embryogenesis and direct/indirect organogenesis were expressed as mean ± standard error. All the experiments were carried out with three replicates, and each experiment was done twice. In the flow cytometric study, three replicates from each regenerated plant group as well as the control were chosen and subjected to flow cytometric analysis. The statistical analyses of data were done with one-way ANOVA using SPSS software (version 15, Chicago, IL, USA). The mean comparisons were determined by DMRT (Duncan’s Multiple Range Test) at *p* < 0.05 [35].

## 3. Results

### 3.1. In Vitro Seed Germination

The in vitro seed germination was successfully obtained on all the three different kinds of MS media (half MS, full MS and full MS with GA_3_). Within 21 days of inoculation, the seedlings started to emerge with the highest frequency in full MS augmented with 0.5 mg/L GA_3_ (73.3%) (Table 1), followed by full-strength MS medium (62.2%) as compared to seeds inoculated in half MS medium (48.9%). The grown seedlings had long hypocotyls in all the three germinating media tested (Figure 1A).

### 3.2. Callus Induction and Proliferation

Three explants (leaf, node and hypocotyl) were cultured in MS having different concentrations as well as combinations of BAP with 2,4-D or NAA. Among these explants, the leaf tissues generated a high incidence (93.1%) of callus formation when a combination of BAP (2.0 mg/L) and NAA (0.5 mg/L) was used (Table 2, Figure 2A). In contrast, the hypocotyl explants showed maximum callusing (90.3%) in MS, amended with 0.75 mg/L BAP, followed by nodal tissues (84.7%) on 2.0 mg/L BAP + 0.5 mg/L 2,4-D. The nodal and leaf produced friable callus of a pale-yellow colour, whereas green and yellow compact calli were developed from hypocotyl explants (Figure 2B).
NOTE: Callus induction frequency (%)=Number of explants showing callusing÷Total number of explants inoculated×100

### 3.3. Somatic Embryo Formation

The subculturing of hypocotyl callus on MS medium added with BAP alone or with 2,4-D/NAA resulted in embryogenic calli within 4–5 weeks of sub-culturing (Figure 3), with a frequency range of 9.7% to 81.9% (Table 3). BAP at 0.75 mg/L showed maximum embryogenesis frequency (81.9 ± 3.7%) with 13.7 ± 2.3 mean numbers of somatic embryos. A gradual decrease in both the frequency (18.1% to 0%) and average count of somatic embryos (2.3 to 0) was observed with increased BAP and 2,4-D (0.5–1.0 mg/L) added medium. Later, the somatic embryos were kept in full MS or with BAP (0.5–2.0 mg/L) for germination. The highest somatic embryo germination (65.3%) was noted on 0.75 mg/L BAP-amended MS medium, which further increased to 76.4% after six weeks of culture (Figure 4). The plantlets grown via somatic embryogenesis were morphologically identical to their parent plantlets.

**Figure 3 genes-13-02335-f003:**
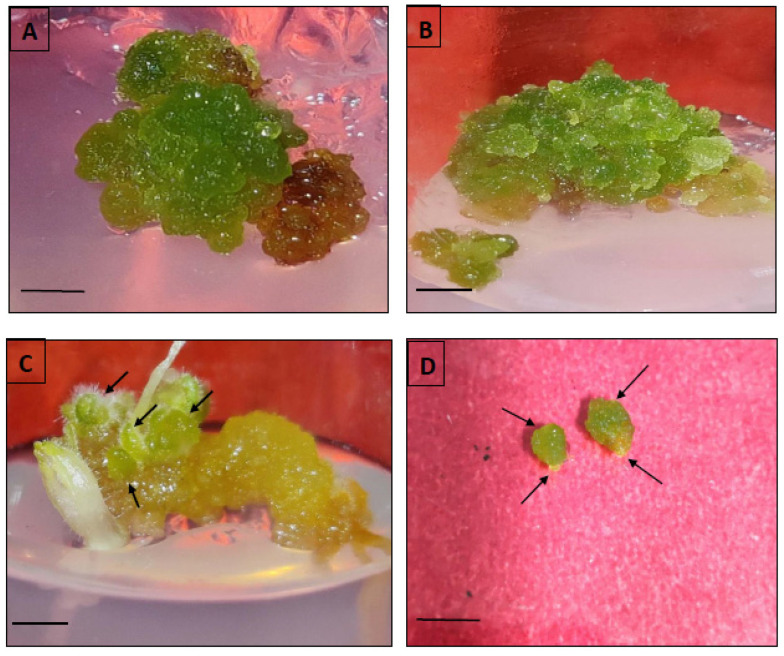
Callus induction and indirect somatic embryogenesis from hypocotyl callus in *D. purpurea*. (**A**,**B**) Callus induction and proliferation (bars (A) = 0.3 cm, (B) = 0.5 cm), (**C**) Somatic embryos formation with different developmental stages (bar = 0.5 cm), (**D**) Isolated bipolar stages of early and late globular embryos with shoot and root ends (arrowheads) (bar = 0.3 cm).

### 3.4. Scanning Electron Microscopy (SEM)

Scanning electron microscopic studies were carried out to determine the origin of somatic embryos from hypocotyl calli. The SEM images revealed the presence of different stages of somatic embryos (globular and heart-shaped) on callus surfaces. The globular embryos were more prominently observed throughout the investigation as compared to the other embryonic stages (Figure 5).

### 3.5. Shoot Organogenesis via Direct and Indirect Method

The shoot regeneration ability of apical buds and cotyledonary nodes cultured on BAP- or kinetin-added MS medium is presented in Table 4. The PGR-free MS medium was ineffective, while incorporation of cytokinins (BAP or Kn) in media led to the induction of shoots in both the two explants. The high incidence of shoot proliferation was observed in apical meristem (93.1%), followed by cotyledonary node (87.5%) on 1.0 mg/L BAP-added MS medium (Figure 1B,C). Relatively low shooting efficiency was noted on kinetin-fortified (0.5 mg/L) MS (11.1% in apical bud and 8.3% in cotyledonary node). The average length of regenerated shoots was 1.9–4.1 cm on BAP-fortified MS medium, whereas in kinetin treatments shoots had a mean length of 1.2–3.4 cm after five weeks of culture (Table 4).

The subculturing of friable callus (leaf- and hypocotyl-derived) on BAP- (0.5–2.0 mg/L) and 2,4-D-added (0.5–1.0 mg/L) MS medium produced successful induction of shoots (indirect) within 5–6 weeks (Figure 2C). The leaf-derived calli were more responsive in developing shoots compared to the hypocotyl callus. Of the several BAP and 2,4-D concentrations examined, 2.0 mg/L BAP + 0.5 mg/L 2,4-D was found to be the best treatment in producing a maximum number of shoots in both the two callus sources (Table 5). The leaf-derived callus showed 70.8% shoot regeneration ability with 10.3 average shoot number/callus mass (Figure 2D), followed by hypocotyls callus having 40.3% shoot induction ability with a 4.3 ± 1.9 average count of shoots after six weeks of culture. At a lower level of BAP (<2 mg/L), a significant reduction in shoot formation was noted in two calli sources. The regenerated shoots were then finally transferred to a medium for root induction.

### 3.6. Rooting and Plantlets Acclimatization

To attain rooting of regenerants, the MS medium was supplied with three different auxins, viz., IBA, IAA and NAA at three concentrations (0.5, 0.75 and 1.0 mg/L) (Figure 6). Except for the control, the roots appeared from the cut ends of the shoots in all the rooting treatments within three weeks. IBA treatments had more impact than the IAA and NAA in terms of root induction percentage and average root numbers per shoot (Figure 7). The highest rooting percentage (87.5%) was recorded on 1.0 mg/L IBA with 24.7 numbers of roots per shoot, whereas the lowest rooting (5.6%) was observed at 0.5 mg/L NAA with 2.3 mean root number/shoot. Those plants with IBA treatments had thicker roots, whereas in IAA and NAA treated plants had narrow and fine roots. About 85–90% of tissue culture raised plants were successfully potted in greenhouse conditions (Figure 7C).

### 3.7. Flow Cytometric Analysis

The determination of ploidy status of in vitro regenerated plants is a prerequisite for establishing true-to-type nature. In the current study, this has been studied by estimating and comparing the nuclear 2C DNA content of in vivo grown and tissue culture raised *D. purpurea* plants. The comparison of flow cytometric histograms showed similar fluorescence peaks (Figure 8), and the estimated 2C DNA contents of in vivo and in vitro raised plants (via somatic embryogenesis and shoot organogenesis) was 3.1, 3.2 and 3.0 pg, respectively (Table 6). No visible changes have been detected between mother and tissue cultured-derived plants, thus confirming genetic stability.

### 3.8. SCoTmarker Analysis

The genetic homogeneity assessment of tissue culture propagated *D. purpurea* with their mother plant was done using SCoT marker (a DNA-based PCR method). Among the 12 SCoT primers tested, six primers amplified 35 and 37 monomorphic bands with a base pair (bp) size ranging from 200–1600 in somatic embryo-regenerated and organogenic-derived plants, respectively (Table 7; Figure 9A,B). An average of six scorable bands per primer was noted. The maximum number (10 bands) of monomorphic bands was produced by SCoT3 primers, while the lowest band number (four) was produced by SCoT33 primer. The banding profiles validated the genetic uniformity of in vitro-derived plantlets with their field-grown mother plants. Dendrogram based on Jaccard’s similarity index of tissue culture-derived and mother plant is presented in Figure 10.

## 4. Discussion

The present investigation aimed to lay down an efficient in vitro plant regeneration protocol via embryogenesis and organogenesis, a prerequisite step to produce elite clonal medicinally important plants which further aids in providing a high yield of pharmaceutically important bio-actives. In the current study, in vitro seed germination, callus induction, direct shoot regeneration, somatic embryogenesis and organogenesis have been successfully conducted in *D. purpurea* by utilizing and optimizing various PGRs (2,4-D, NAA, BAP, kinetin). Maximum in vitro seed germination of *D. purpurea* was observed in 0.5 mg/L GA_3_ amended full MS medium, and the germinated seedlings were found to possess longer hypocotyls. Seed dormancy is a common problem encountered in various plant species that leads to delayed fruit set formation and reduced production of viable plants. GA_3_ is widely known to break seed dormancy, and it has also been used in promoting the seed germination rate under in vitro conditions in other species of *Digitalis*, such as *D. ferruginea* [37] and *D. cariensis* [8].

For direct shoot regeneration, the apical meristems and cotyledonary nodal parts of in vitro grown seedlings were then cultured onto shoot proliferation medium. The cytokinins BAP and kinetin were proved to be effective in inducing shoots in both the explants in *D. purpurea*. However, the employment of BAP at 1.0 mg/L was more productive in shoot proliferation as compared to other BAP treatments. An increase or decrease in BAP level lowered shoot induction rate and shoot length. Similar results have also been described in different plant species [19,38]). On the other hand, the incorporation of kinetin showed moderate shoot growth in both the explants with the highest shooting frequency of 58.3%. Thus, in our study, BAP was found to be a superior cytokinin in regenerating shoots than kinetin. Earlier reports also suggested that BAP is a better cytokinin for shoot regeneration in several plants, such as *Chenopodium quinoa* [39], *Eurycomalongifolia* [40] and *Cicer arietinum* [41].

Callogenesis has been proved to be a valuable tool to obtain in vitro regenerated plants via embryogenesis/organogenesis. In *Digitalis*, callus induction was achieved from different explants in several species [42,43,44]. Here, in this study, all the tested explants (leaf, node and hypocotyls) were able to induce callus in BAP-amended MS medium with or without 2,4-D/NAA at a concentration level of 0.5–2.0 mg/L. The obtained results are in accordance with earlier findings reported by Rad et al. [4] and Mamgain et al. [45]. The organogenic potential was also examined wherein the leaf and hypocotyls-derived calli produced shoots in BAP- and 2,4-D-amended MS medium at different concentrations. These findings are in good agreement with Flores et al. [46], who found that the combination of BAP and 2,4-D was found to be proficient in inducing shoot organogenesis. In the current study, cytokinin at higher concentrations stimulated shoot-bud formation from callus more efficiently than auxin. The results are in accordance with Hesami and Dhaneshvar [47], who found that 1.5 mg/L BAP along with 0.15 mg/L IBA showed 100% shoot regeneration frequency. Contrary to our observation, BAP with NAA was proved to be an efficient combination for indirect shoot regeneration in *Digitalis ferruginea* [37], *Rauvolfia serpentina* [48] and *Aspilia africana* [49].

Within five weeks, embryogenic calli were obtained from hypocotyl callus in the same medium. Generally, the auxins (2,4-D, IAA, NAA) alone or in combination with cytokinins (BAP, Kn, TDZ) are widely known to induce somatic embryos (directly or indirectly) and are noted in several plants. In our study, BAP alone was more effective in somatic embryo development (also in combination with auxins). However, the mean number of somatic embryos developed per callus was quite low in different tested treatments. Our observations are very similar and consistent with previous reports, investigating *Chrysanthemum* spp. [50] and *Metabriggsia ovalifolia* [51]. SEM analyses confirmed the origin of somatic embryos and demonstrated the progression through different developmental stages (globular, heart) of embryogenesis. The successful conversion of somatic embryos into healthy plantlets was achieved more efficiently in MS fortified with varied levels of BAP as compared to PGR-free MS medium. Similar somatic embryo germination responses were recorded in PGR-added media in various plant species [52,53]. Somatic embryo germination is, however, frequently regulated by a number of factors including plant genotype, PGRs, culture conditions, photoperiod, and in interaction with endogenous levels of hormones [54]. The root-induction ability of tissue culture raised plants was tested with three different auxins, namely IAA, IBA and NAA. Among those, IBA at 1.0 mg/L exhibited the best response in producing roots in comparison to the other two auxins i.e., IAA and NAA. Sinchana et al. [55] attributed this root-promoting phenomenon towards efficient absorption and metabolisation of IBA by the cultured shoots. Similar rooting ability with IBA has been reported in several plants such as *Curcuma zedoaria* [56] and *Solanum khasianum* [57].

Plantlets obtained via an intervening callusing stage (embryogenesis/organogenesis) are a good source of somaclonal variation showing ploidy level changes, epigenetic variations and other mutations [15]. To ensure true-to-type plant propagation, flow cytometry and SCoT marker-based PCR techniques have been attempted and showed genetic homogeneity with field-grown mother plants. The similarity in histogram peaks confirmed no genetic variation in laboratory-grown plants. Similarly, the 2C nuclear DNA was estimated recently in plants like *Curcuma angustifolia* [58], *Cucumis melo* [59], *Disanthus cercidifolius* [60] and in other species of *Digitalis*, i.e., *D. lanata* [30].

The genetic uniformity of *Digitalis* was further confirmed by SCoT-PCR analysis, and this is the first report where genetic homogeneity was assessed in *D. purpurea* by using SCoT. Here, six different SCoT primers (SCoT 3, SCoT 7, SCoT 14, SCoT 16, SCoT 26 and SCoT 33) were employed and all in vitro raised plants (randomly selected) produced monomorphic bands revealing 94.3% similarity in somatic embryo-derived plants and 95.13% similarity in organogenic-derived plantlets with the parent plant. In recent years, SCoT marker-based molecular analysis of genetic uniformity of in vitro grown plants has been conducted in multiple plant varieties [61,62,63,64]. Perez-Alonso et al. [9] investigated genetic homogeneity of *D. purpurea* micropropagated plants by using RAPD analyses earlier. These results suggest that SCoT marker-assisted genetic evaluation of tissue culture raised plants can also be applied in other *Digitalis* species.

## 5. Conclusions

The present investigation demonstrates an efficient in vitro regeneration protocol of *Digitalis purpurea* for mass propagation of plantlets which can be exploited pharmaceutically for medicinal uses. Flow cytometry and SCoT marker analysis clearly confirmed the genetic uniformity of regenerated populations. Due to greater applicability, reliability and reproducibility, flow cytometry and SCoT marker-based approaches have proved to be robust techniques in detecting genetic homogeneity/somaclonal variations of regenerated plants with high commercial values.

## Figures and Tables

**Figure 1 genes-13-02335-f001:**
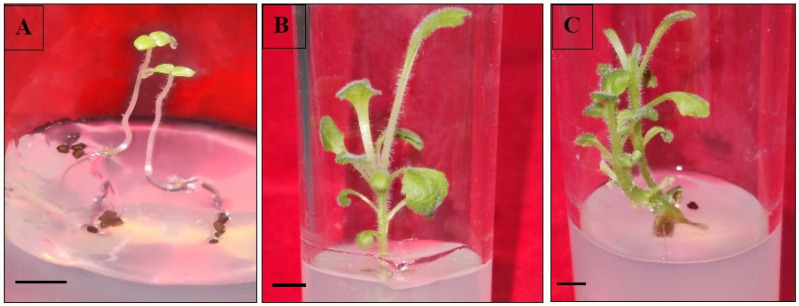
In vitro germination of seeds and direct shoot regeneration in *D. purpurea*. (**A**) Germinated seedlings having elongated hypocotyls (bar = 0.5 cm). (**B**,**C**) In vitro direct shoot proliferation (bars = 1.0 cm).

**Figure 2 genes-13-02335-f002:**
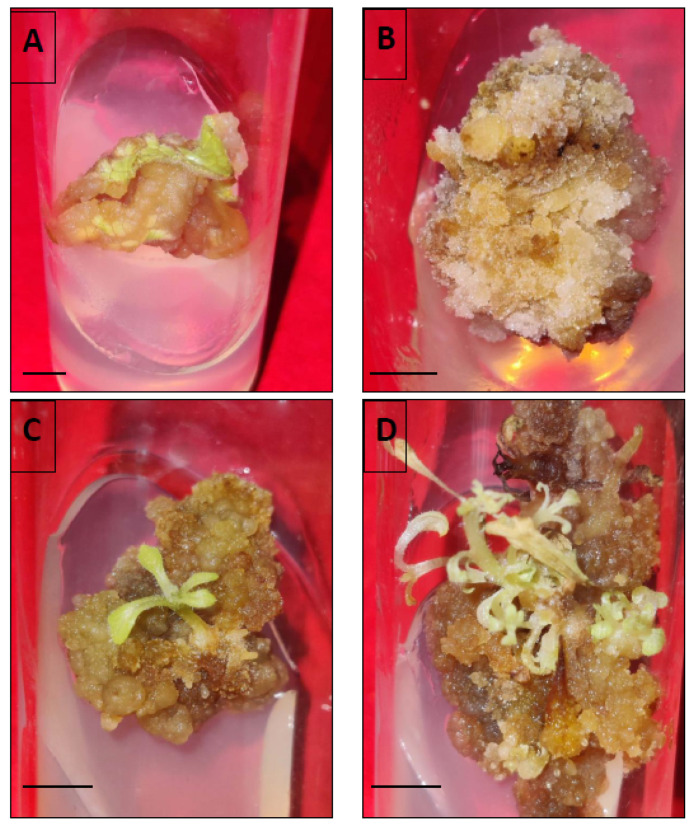
Callus induction and callus-mediated shoot organogenesis from leaf explant in *D. purpurea*. (**A**) Callus initiation (bar = 1.0 cm), (**B**) Callus proliferation after postweeks (bar = 0.5 cm), (**C**,**D**) Shoot regeneration from leaf derived callus (bars = 0.5 cm).

**Figure 4 genes-13-02335-f004:**
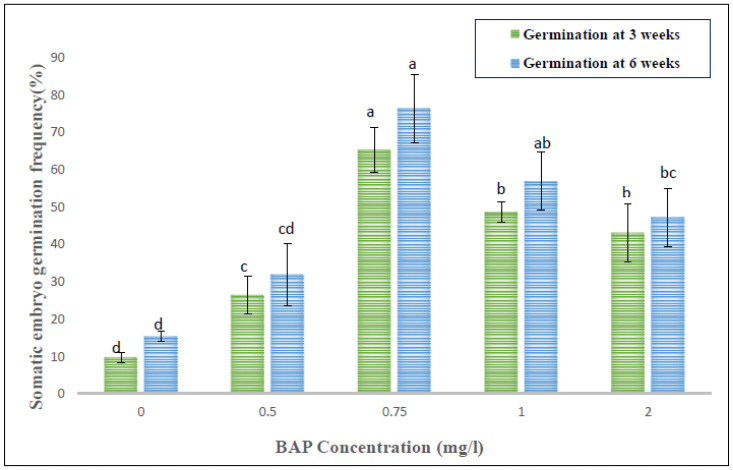
Effect of BAP-fortified MS medium on somatic embryo germination after a three- and six-week period, respectively. Values indicate the Mean ± SE of three replicates per treatment. Mean values with different superscripts within a column are significantly different from each other as per DMRT at *p* ≤ 0.05.

**Figure 5 genes-13-02335-f005:**
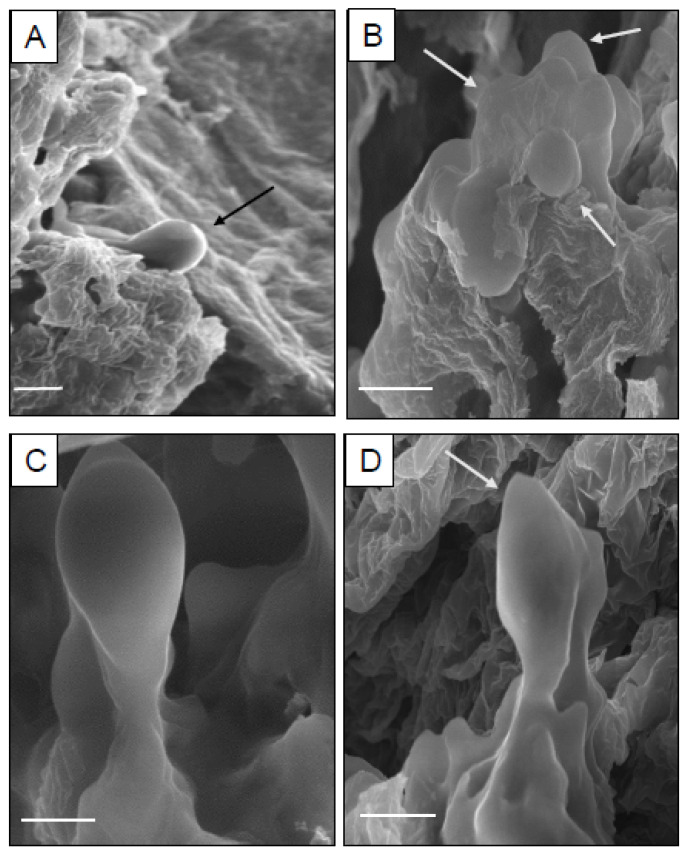
Scanning electron microscopic (SEM) images of somatic embryos in *D. purpurea*. (**A**) Globular shaped somatic embryo (bar = 20 μm), (**B**) Cluster of globular shaped somatic embryos (bar = 2 μm), (**C**) Globular shaped embryo with suspensor (bar = 2 μm), (**D**) Heart shaped somatic embryo (bar = 2 μm).

**Figure 6 genes-13-02335-f006:**
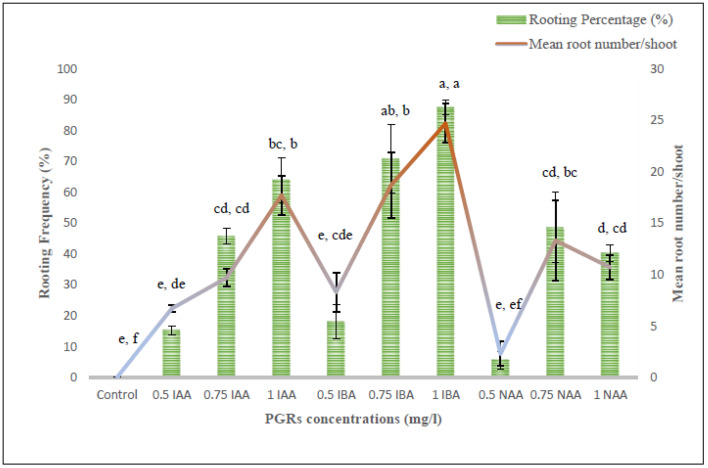
Effect of various PGRs concentration on root induction frequency and mean number of roots/shoot. Values indicate the mean ± SE of 3 replicates per treatment. Mean values with different superscripts within a column are significantly different from each other as per DMRT at *p* ≤ 0.05.

**Figure 7 genes-13-02335-f007:**
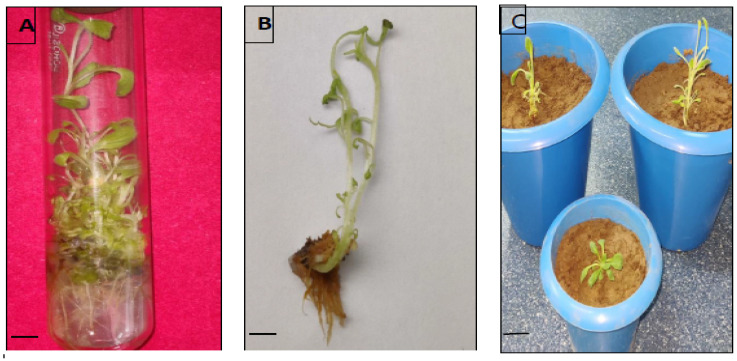
(**A**,**B**) Root induction of in vitro regenerated *D. purpurea* plants (bars = 1.0 cm). (**C**) Tissue culture raised *D. purpurea* plants in pots (bar = 3 cm).

**Figure 8 genes-13-02335-f008:**
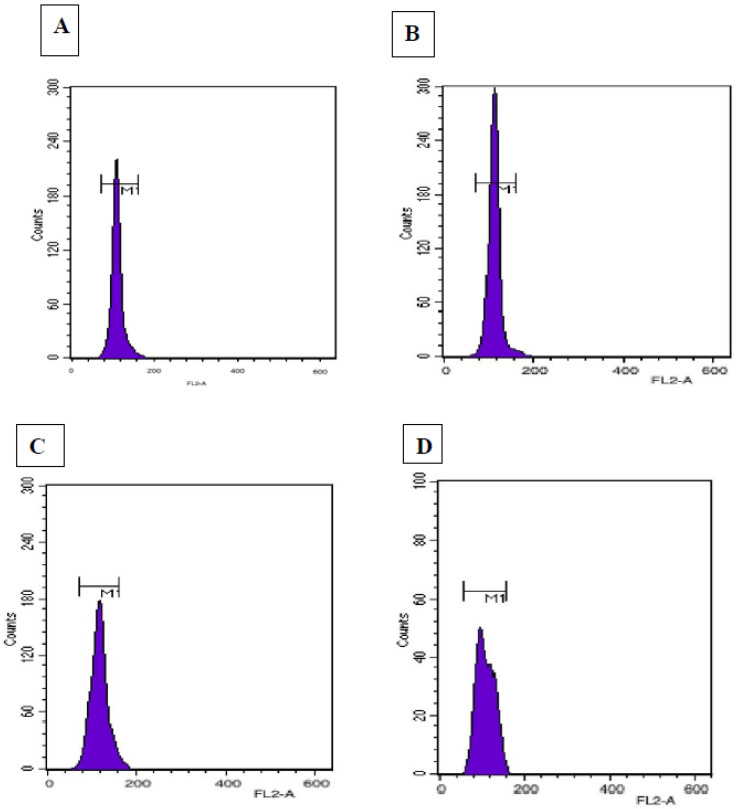
Flow cytometric histogram peaks of relative 2C DNA content of nuclei obtained from (**A**) *Digitalis lanata* (standard), (**B**) Field-grown (mother) plant, (**C**) somatic embryo regenerated plant, and (**D**) organogenic-derived plant of *Digitalis purpurea*.

**Figure 9 genes-13-02335-f009:**
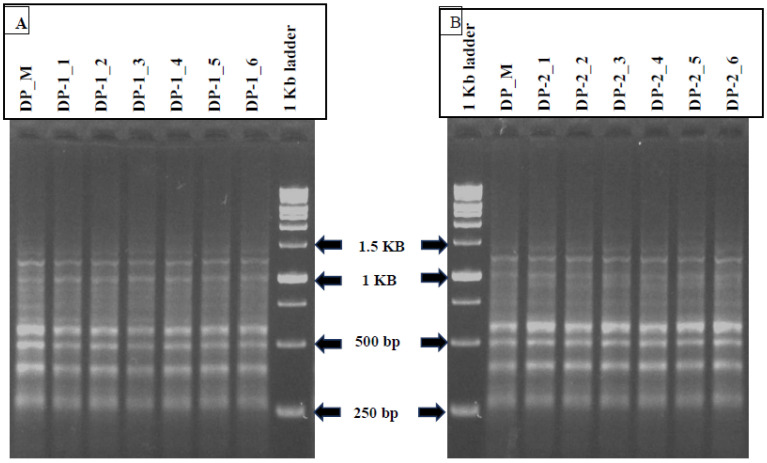
SCoT banding profiles of in vitro propagated plants with mother plant (DP_M) of *D. purpurea* with SCoT 14 primer. (**A**) DP-1_1 to DP-1_6 (somatic embryo-derived plants), (**B**) DP-2_1 to DP-2_6 (organogenic-derived plants).

**Figure 10 genes-13-02335-f010:**
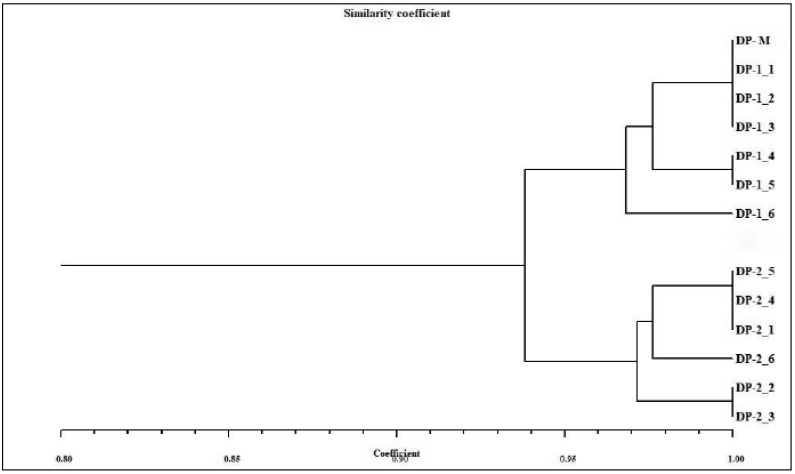
Dendrogram of SCoT analysis depicting the genetic similarity between the mother plant (DP-M) and the in vitro regenerated plants of *D. purpurea*.

**Table 1 genes-13-02335-t001:** In vitro seed germination frequency of *Digitalis purpurea* after three weeks.

Medium	Germination Rate (%)
Half MS	48.9 ± 4 ^b^
Full MS	62.2 ± 1.1 ^a^
Full MS + 0.5 mg/L GA_3_	73.3 ± 3.8 ^a^

Values indicate mean of 25 seeds inoculated in different media. Total seeds observed (25 × 3 = 75). Germinate Rate % = No. of germinated seedlings/Total numbers of seeds inoculated × 100. Mean values with the different superscripts within a column are significantly different from each other as per DMRT at *p* ≤ 0.05.

**Table 2 genes-13-02335-t002:** Effects of different concentrations as well as combinations of PGRs on callus induction frequency from hypocotyl, node and leaf in *D. purpurea*.

PGRs	Concentration (mg/L)	Explants		
		Hypocotyl	Node	Leaf
Control	0	0 ^g^	0 ^i^	0 ^i^
BAP	0.5	51.4 ± 10.8 ^c^	9.7 ± 2.8 ^h,i^	12.5 ± 2.4 ^f,g,h,i^
	0.75	90.3 ± 1.4 ^a^	19.4 ± 3.7 ^g,h^	20.8 ± 4.2 ^e,f,g,h^
	1	58.3 ± 4.2 ^b,c^	11.1 ± 1.4 ^h,i^	6.9 ± 3.7 ^g,h,i^
	2	30.6 ± 2.8 ^d,e,f^	12.5 ± 2.4 ^h,i^	5.6 ± 1.4 ^h,i^
BAP + 2,4-D	0.5 + 1.0	18.1 ± 6.1 ^f,g^	22.2 ± 3.7 ^g,h^	13.9 ± 3.7 ^f,g,h,i^
	0.5 + 2.0	40.3 ± 10 ^c,d,e^	26.4 ± 3.7 ^f,g,h^	25 ± 6.4 ^d,e,f,g,h^
	1.0 + 0.5	45.8 ± 13.4 ^c,d^	36.1 ± 5 ^f,g^	29.2 ± 2.4 ^d,e,f^
	1.0 + 2.0	50 ± 10.5 ^c^	43.1 ± 5 ^e,f^	37.5 ± 4.8 ^c,d,e^
	2.0 + 0.5	0 ± 0 ^g^	84.7 ± 6.9 ^a^	77.8 ± 6.1 ^a^
	2.0 + 1.0	0 ± 0 ^g^	58.3 ± 4.2 ^c,d,e^	54.2 ± 10.5 ^b,c^
BAP + NAA	0.5 + 1.0	25 ± 2.4 ^e,f^	54.2 ± 6.4 ^d,e^	26.4 ± 1.4 ^d,e,f,g^
	0.5 + 2.0	70.8 ± 4.2 ^b^	56.9 ± 9.7 ^c,d,e^	43.1 ± 10.8 ^b,c,d^
	1.0 + 0.5	12.5 ± 2.4 ^f,g^	68.1 ± 7.3 ^a,b,c,d^	29.2 ± 4.8 ^d,e,f^
	1.0 + 2.0	0 ± 0 ^g^	65.3 ± 10.8 ^b,c,d^	59.7 ± 13.2 ^b^
	2.0 + 0.5	0 ± 0 ^g^	73.6 ± 7.3 ^a,b,c^	93.1 ± 1.4 ^a^
	2.0 + 1.0	0 ± 0 ^g^	81.9 ± 1.4 ^a,b^	84.7 ± 5 ^a^

Values indicate means ± standard errors of three replicates of two experiments. Mean values with different superscripts within a column are significantly different from each other as per DMRT at *p* ≤ 0.05. The superscripts denote the ranking as per Duncan test i.e ‘a’ represents the best treatment and ‘i’ represents the lowest value.

**Table 3 genes-13-02335-t003:** Effect of different concentrations as well as combinations of PGRs on somatic embryogenesis from hypocotyl callus in *D. purpurea*.

PGRs	Concentration (mg/L)	Frequency (%) of Somatic Embryogenesis	Mean Number of SEs/Callus Mass (250 mg)
Control	0	0 ^g^	0 ^f^
BAP	0.5	56.9 ± 7.3 ^b^	11.3 ± 0.9 ^a,b^
	0.75	81.9 ± 3.7 ^a^	13.7 ± 2.3 ^a^
	1.0	45.8 ± 6.4 ^b,c^	8.3 ± 1.9 ^b,c^
BAP + 2,4-D	0.5 + 1.0	18.1 ± 3.7 ^e,f^	2.3 ± 0.3 ^d,e,f^
	0.75 + 0.5	9.7 ± 2.8 ^f,g^	0.7 ± 0.3 ^e,f^
	1.0 + 0.5	0 ± 0 ^g^	0 ± 0 ^f^
BAP + NAA	0.5 + 1.0	34.7 ± 9.7 ^c,d^	5.7 ± 1.5 ^c,d^
	0.75 + 0.5	55.6 ± 3.7 ^b^	11.7 ± 1.3 ^a,b^
	1.0 + 0.5	26.4 ± 3.7 ^d,e^	4.3 ± 0.9 ^d,e^

Values indicate means ± standard errors of three replicates of two experiments. Mean values with different superscripts within a column are significantly different from each other as per DMRT at *p* ≤ 0.05. SEs: Somatic Embryos. The superscripts denote the ranking as per Duncan test i.e ‘a’ represents the best treatment and ‘i’ represents the lowest value.

**Table 4 genes-13-02335-t004:** Effect of BAP and Kn on direct shoot regeneration in *D. purpurea.*.

PGRs	Concentration (mg/L)	Shoot Proliferation Frequency (%)		Mean No. of shoots/explant		Length of Shoots (cm)	
		Apical Bud	Cotyledonary Node	Apical Bud	Cotyledonary Node	Apical Bud	Cotyledonary Node
Control	0	0 ^f^	0 ^e^	0 ^d^	0 ^e^	0 ^e^	0^e^
BAP	0.5	45.8 ± 4.8 ^c,d^	18.1 ± 6.1 ^d^	4.7 ± 0.9 ^ab^	2.7 ± 0.3 ^bc^	3.7 ± 0.3 ^a^	2.1 ± 0.2 ^b,c^
	0.75	77.8 ± 5.6 ^b^	81.9 ± 6.9 ^a^	5.3 ± 0.3 ^a^	4.7 ± 0.9 ^ab^	3.0 ± 0.5 ^a,b^	3.7 ± 0.2 ^a^
	1.0	93.1 ± 1.4 ^a^	87.5 ± 2.4 ^a^	5.3 ± 0.7 ^a^	5.7 ± 0.9 ^a^	3.8 ± 0.4 ^a^	4.1 ± 0.3 ^a^
	2.0	26.4 ± 1.4 ^e^	37.5 ± 2.4 ^c^	3.3 ± 0.3 ^bc^	3.7 ± 0.7 ^abc^	1.9 ± 0.1 ^c,d^	3.8 ± 0.2 ^a^
Kn	0.5	11.1 ± 1.4 ^f^	8.3 ± 4.2 ^d,e^	2.0 ± 1.2 ^c^	0.3 ± 0.3 ^de^	1.2 ± 0.2 ^d^	1.2 ± 0.2 ^d^
	0.75	58.3 ± 8.7 ^c^	44.4 ± 5 ^b,c^	4.3 ± 0.3 ^ab^	4.3 ± 0.9 ^abc^	2.2 ± 0.2 ^b,c^	2.3 ± 0.3 ^b,c^
	1.0	34.7 ± 5 ^d,e^	58.3 ± 4.2 ^b^	3.7 ± 0.3 ^abc^	5.0 ± 0.6 ^a^	3.4 ± 0.3 ^a^	2.7 ± 0.3 ^b^
	2.0	29.2 ± 2.4 ^e^	20.8 ± 6.4 ^d^	2.3 ± 0.3 ^c^	2.3 ± 0.9 ^cd^	2.0 ± 0.2 ^c,d^	2.0 ± 0.1 ^c^

Values indicate means ± standard errors of three replicates of two experiments. Mean values with different superscripts within a column are significantly different from each other as per DMRT at *p* ≤ 0.05. The superscripts denote the ranking as per Duncan test i.e ‘a’ represents the best treatment and ‘i’ represents the lowest value.

**Table 5 genes-13-02335-t005:** Effect of BAP and 2,4-D on shoot organogenesis from leaf and hypocotyl callus in *D. purpurea*.

PGRs	Concentration (mg/L)	Frequency of Organogenesis (%)		Mean No. of Shoots/Callus Mass	
		Leaf	Hypocotyl	Leaf	Hypocotyl
Control	0	0 ^d^	0 ^c^	0 ^e^	0 ^b^
BAP + 2,4-D	0.5 +1.0	18.1 ± 6.1 ^c,d^	8.3 ± 4.2 ^c^	2.3 ± 0.3 ^d^	2.3 ± 1.2 ^a,b^
	0.75 + 1.0	43.1 ± 10 ^b^	6.9 ± 1.4 ^c^	4.3 ± 0.7 ^c,d^	1.3 ± 0.3 ^a,b^
	1.0 + 0.5	56.9 ± 3.7 ^a,b^	22.2 ± 1.4 ^b^	7.3 ± 0.9 ^b^	4.3 ± 1.9 ^a^
	1.5 + 0.75	37.5 ± 4.8 ^b,c^	29.2 ± 2.4 ^b^	4.7 ± 0.9 ^c^	2.3 ± 0.9 ^a,b^
	2.0 + 0.5	70.8 ± 8.7 ^a^	40.3 ± 3.7 ^a^	10.3 ± 0.9 ^a^	3.7 ± 1.2 ^a,b^

Values indicate means ± standard errors of three replicates of two experiments. Mean values with different superscripts within a column are significantly different from each other as per DMRT at *p* ≤ 0.05. The superscripts denote the ranking as per Duncan test i.e ‘a’ represents the best treatment and ‘i’ represents the lowest value.

**Table 6 genes-13-02335-t006:** Comparison of nuclear 2C DNA content and genome size of in vitro propagated plants (somatic embryo regenerated and callus-mediated plants) and donor plants of *D. purpurea*.

Plant Sample Types	2C DNA Content (pg *)	1C DNA Content (pg)	Genome Size (Mbp)
In vivo (donor) plant	3.1 ± 0.01 ^a^	1.55 ± 0.0 ^a^	3031.8 ^a^
Somatic embryo regenerated plant	3.2± 0.01 ^a^	1.6 ± 0.01 ^a^	3129.6 ^a^
Callus-mediated plant	3.0 ± 0.01 ^a^	1.5 ± 0.0 ^a^	2934 ^a^

Values indicate means ± standard errors of three replicates of two experiments. Mean values with different superscripts within a column are significantly different from each other as per DMRT at *p* ≤ 0.05. * 1 pg = 978 Mbp as per Dolezel [36].

**Table 7 genes-13-02335-t007:** List of SCoT primers, their sequences, %G/C, Tm, number of bands and their approximate band length (bp) obtained in micropropagated plantlets of *D. purpurea*.

S. No	Primer Name	Primer Sequences (5′-3′)	%G/C	Tm (°C)	No. of Bands Amplified		Approximate Band Length (bp)
					DP_1 *	DP_2 *	
1	SCoT3	CAACAATGGCTACCACCG	56	48 °C	10	10	300–1400
2	SCoT7	CAACAATGGCTACCACGG	56	48 °C	5	5	300–1500
3	SCoT14	ACGACATGGCGACCACGC	56	48 °C	6	6	300–1250
4	SCoT16	ACCATGGCTACCACCGAC	56	48 °C	7	6	300–1400
5	SCoT26	ACCATGGCTACCACCGTC	61	50 °C	5	4	500–1500
6	SCoT33	CCATGGCTACCACCGCAG	67	50 °C	4	4	200–1600

* DP_1 and DP_2 represent somatic embryo regenerated plantlets and callus-mediated plantlets, respectively.

## Data Availability

Not applicable.

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
