# Peer review of "Ploidy Status, Nuclear DNA Content and Start Codon Targeted (SCoT) Genetic Homogeneity Assessment in Digitalis purpurea L., Regenerated In Vitro"

_genes, 2022, doi:10.3390/genes13122335_

Round 1
Reviewer 1 Report
I am curious;
Why three replicates and two experiments??
Why not six replicates per experiment??

Author Response
Reviewer 1
- In the experimental design: It means that, for example, for each concentration of PGR, he only used three tubes. used three leaf fragments, three hypocotyl fragments, etc. And all these experiments were performed twice?? Why not each experiment with six repetitions?
Ans: In plant tissue culture, a number of plant growth regulators and their combinations are tested using different explants like hypocotyl, leaf etc. We used a minimum of three replica and the treatments were performed twice i.e. six repetitions (3x2=6, as the reviewer indicated). For example, leaf is the explant, 2,4-D is the PGR and 0.5 mg/l is the concentration (PGR, concentration and explant, there are three variables) which produce a big sample size. Higher repetitions like 8, 10 or more will produce even bigger sample size. Lowering the size instead reduces experimental errors, eases subculturing pressure, and minimizes cost and labour. Owing to several advantages, replications are kept in check and are widely used practice in laboratories.
- Does not coincide with image A in figure 1
Ans: The above is now corrected.
Reviewer 2 Report
Materials and Methods
- Please describe the measurement method of callus induction frequency in 2.1 (p 3 Line 112-119) and also specify those of terms as note in Table 2 (p6).
Results
- Specify what the arrows in Figure 3 and 4 indicate.
Author Response
Reviewer 2
- Please describe the measurement method of callus induction frequency in 2.1 (p 3 Line 112-119) and also specify those of terms as note in Table 2 (p6).
Ans: The measurement method has been described.
- Specify what the arrows in Figure 3 and 4 indicate.
Ans: Specified.
Reviewer 3 Report
The study is interesting and generally well-designed. Still, some issues need to be clarified and improved, which I detail below.
In all the statistical analyzes where ANOVA was performed from percentage data, it is necessary to transform the data and perform some test of homoscedasticity (homogeneity of variances) to be valid; if said homogeneity is not found, tests will have to be performed parametric.
The evidence of the formation of somatic embryos is not very clear; the type of callus shown in figure 3 does not show the general characteristics of an embryogenic callus and the morphogenic response could be indirect organogenesis, they also mention that 13.7 ± 2.3 mean numbers were recorded of somatic embryos, it would be convenient to show the evidence of this average in a photo, since the photos in figure 3 only show a few morphogenic structures. The SEM photographs also do not clearly show the formation of ES in the stages that are mentioned, the authors need to offer reliable evidence of the formation of somatic embryos, it is necessary to make histological sections that show the characteristic structures of an ES, that is, the bipolar development of meristems and the formation of vascular bundles independent of the original tissue.
It will be helpful if the authors show images of the germination process.
The authors show in table 4 the Frequency of Shoot Proliferation (%), and it would be advisable to indicate the number of shoots per explant (apical buds and cotyledonary nodes), as they do in table 5, to evaluate the efficiency in the formation of plants, since if the response to bud formation is as shown in Figure 2C, the in vitro propagation system may not be as efficient as they stated.
It would be very illustrative to show photos of those callus that formed up to around ten shoots per leaf-derived callus, as well as the process of development, rooting of plants, and their establishment in the greenhouse.
The authors used different terms to describe the maintenance of the genetic constitution of a particular clone: genetic integrity, genetic similarity, genetic fidelity, genetic homogeneity, and they can cause confusion since some of them are used to describe different situations, from clone, population and species, use the term that best suits the study.
What treatment do the flow cytometry and ScoTT study samples come from?
In the discussion section, it is correct to mention that certain PGRs favor the organogenic response, but it is also known that specific concentrations of these PGRs significantly influence the different responses, the authors could discuss a little more in this regard.
En los análisis estadísticos donde se realizó ANOVA a partir de datos porcentuales, es necesario la transformación de los datos y realizar alguna prueba de homocedasticidad (homogeneidad de varianzas) para que sea válida, si no se encuentra dicha homogeneidad, se tendrá que realizar pruebas no paramétricas.
There is yet no clear evidence for SE induction and, as the authors point out in the discussion, embryogenesis has been recorded preferentially in the presence of some auxin and rarely, but not impossible, in media supplemented with cytokinin.
There is no photographic evidence of the entire micropropagation process, and the current results and figures do not show such a rapid and repeatable in vitro regeneration protocol of D. puruprea “The present investigation demonstrates a rapid and repeatable in vitro regeneration protocol of Digitalis purpurea for large scale mass propagation of plantlets”
In the discussion about organogenesis and rooting, nothing is mentioned and discussed regarding the effect of different concentrations used in the experiment and its comparison with other studies.
Author Response
- In all the statistical analyzes where ANOVA was performed from percentage data, it is necessary to transform the data and perform some test of homoscedasticity (homogeneity of variances) to be valid; if said homogeneity is not found, tests will have to be performed parametric.
Ans: The concept is not clear. Genetic homogeneity or somaclonal variation is a simple phenomenon. Flowcytometry and SCoT marker confirmed that there is no variation in culture regenerated plants. We are afraid about the relevance of other unnecessary tests as the selection of plants was made randomly and the molecular / DNA test analysis was made accordingly. A large number of papers have been published recently from our laboratory following the same statistical method, described in this article.
- The evidence of the formation of somatic embryos is not very clear; the type of callus shown in figure 3 does not show the general characteristics of an embryogenic callus and the morphogenic response could be indirect organogenesis, they also mention that 13.7 ± 2.3 mean numbers were recorded of somatic embryos, it would be convenient to show the evidence of this average in a photo, since the photos in figure 3 only show a few morphogenic structures. The SEM photographs also do not clearly show the formation of ES in the stages that are mentioned, the authors need to offer reliable evidence of the formation of somatic embryos, it is necessary to make histological sections that show the characteristic structures of an ES, that is, the bipolar development of meristems and the formation of vascular bundles independent of the original tissue.
Ans: The early stages of somatic embryos are clearly visible with naked eyes on callus surfaces. A photo panel is now included and added (Fig. 3D) which shows distinct bipolar embryos with green shoot end and rudimentary root axis (arrow head). This morphological appearance is good enough proof of early-stage embryo to scientists, working with embryogenesis. No other evidence is really needed to back up the case. We however, provided SEM images in further confirming embryogenesis. The morphology of bipolar embryo and SEM images all clearly established embryogenesis mediated plant regeneration.
- It will be helpful if the authors show images of the germination process.
Ans: The image of seed germination has already been in place, shown in Figure 1A wherein the emergence of cotyledonary leaves marks the beginning of this process.
- The authors show in table 4 the Frequency of Shoot Proliferation (%), and it would be advisable to indicate the number of shoots per explant (apical buds and cotyledonary nodes), as they do in table 5, to evaluate the efficiency in the formation of plants, since if the response to bud formation is as shown in Figure 2C, the in vitro propagation system may not be as efficient as they stated.
Ans: The data has been added in table 4.
- It would be very illustrative to show photos of those calluses that formed up to around ten shoots per leaf-derived callus, as well as the process of development, rooting of plants, and their establishment in the greenhouse.
Ans: The required photos have been added (Fig. 7C) in revised photoplate.
- The authors used different terms to describe the maintenance of the genetic constitution of a particular clone: genetic integrity, genetic similarity, genetic fidelity, genetic homogeneity, and they can cause confusion since some of them are used to describe different situations, from clone, population and species, use the term that best suits the study.
Ans: All the above terminology is more or less the same meaning; we now used the right word (homogeinity) as per reviewer suggestion.
- What treatment do the flow cytometry and ScoT study samples come from?
Ans: Flow cytometry and SCoT were used to study genetic homogeneity of tissue culture regenerated plants by comparing mother or parent plants. The plants were selected randomly from these two sources (in vitro and field grown). Please see the material and method (section 2.8) and table 6 for more detail. We are afraid, no treatment is necessary.
- In the discussion section, it is correct to mention that certain PGRs favor the organogenic response, but it is also known that specific concentrations of these PGRs significantly influence the different responses, the authors could discuss a little more in this regard.
Ans: This suggestion has been incorporated in the manuscript.
- There is yet no clear evidence for SE induction and, as the authors point out in the discussion, embryogenesis has been recorded preferentially in the presence of some auxin and rarely, but not impossible, in media supplemented with cytokinin.
Ans: In discussion part, we mentioned that the auxins in general (singly or in combination with cytokinins) induce somatic embryos in culture. But in this present study, BAP – a cytokinin alone was more effective in inducing somatic embryos (also in combination with with auxins).The text is now bit modified.
- There is no photographic evidence of the entire micropropagation process, and the current results and figures do not show such a rapid and repeatable in vitro regeneration protocol of puruprea. “The present investigation demonstrates a rapid and repeatable in vitro regeneration protocol of Digitalis purpurea for large scale mass propagation of plantlets”
Ans: In this present study, authors’ primary focus was on to investigate the ploidy status, nuclear DNA content and start codon targeted (SCoT) genetic homogeneity in regenerated Digitalis purpurea (not micropropagation as such). Please see the title of paper. This is the reason micropropagation related discussion (reviewer#3 mentioned in report) is not elaborated and avoided in manuscript. Many useful information has however, been automatically appeared as some of these steps are unavoidable (if not all). Here, in this article, the seed germination process, callus formation from explants, shoot induction and root regeneration steps have been discussed. Our primary job is little different, still we covered and provided photoplates of micropropagation stages (starting from seed germination to outdoor plant transfer through shoot and root induction).The photographic evidence of plants transfer to outdoor (which was omitted earlier) is now added as per reviewer suggestion.
- In the discussion about organogenesis and rooting, nothing is mentioned and discussed regarding the effect of different concentrations used in the experiment and its comparison with other studies.
Ans: As per suggestion little modification in text is made.
Round 2
Reviewer 3 Report
- In all the statistical analyzes where ANOVA was performed from percentage data, it is necessary to transform the data and perform some test of homoscedasticity (homogeneity of variances) to be valid; if said homogeneity is not found, tests will have to be performed parametric.
Ans: The concept is not clear. Genetic homogeneity or somaclonal variation is a simple phenomenon. Flowcytometry and SCoT marker confirmed that there is no variation in culture regenerated plants. We are afraid about the relevance of other unnecessary tests as the selection of plants was made randomly and the molecular / DNA test analysis was made accordingly. A large number of papers have been published recently from our laboratory following the same statistical method, described in this article.
This is basic statistics, it is necessary to seek advice from a specialist in statistics, and I consider that it is not a valid answer that the authors mention " A large number of papers have been published recently from our laboratory following the same statistical method," I emphasize, this is basic statistics.
- The evidence of the formation of somatic embryos is not very clear; the type of callus shown in figure 3 does not show the general characteristics of an embryogenic callus and the morphogenic response could be indirect organogenesis, they also mention that 13.7 ± 2.3 mean numbers were recorded of somatic embryos, it would be convenient to show the evidence of this average in a photo, since the photos in figure 3 only show a few morphogenic structures. The SEM photographs also do not clearly show the formation of ES in the stages that are mentioned, the authors need to offer reliable evidence of the formation of somatic embryos, it is necessary to make histological sections that show the characteristic structures of an ES, that is, the bipolar development of meristems and the formation of vascular bundles independent of the original tissue.
Ans: The early stages of somatic embryos are clearly visible with naked eyes on callus surfaces. A photo panel is now included and added (Fig. 3D) which shows distinct bipolar embryos with green shoot end and rudimentary root axis (arrow head). This morphological appearance is good enough proof of early-stage embryo to scientists, working with embryogenesis. No other evidence is really needed to back up the case. We however, provided SEM images in further confirming embryogenesis. The morphology of bipolar embryo and SEM images all clearly established embryogenesis mediated plant regeneration.
I stand by all the comments that I have already made, and despite the fact that some more developed morphogenetic structures could be considered or confused as somatic embryos in figure 3 C-D, it is necessary to provide irrefutable evidence of the formation of the ES in its different stages.
- It will be helpful if the authors show images of the germination process.
Ans: The image of seed germination has already been in place, shown in Figure 1A wherein the emergence of cotyledonary leaves marks the beginning of this process.
I was referring to the germination of somatic embryos process.
- There is no photographic evidence of the entire micropropagation process, and the current results and figures do not show such a rapid and repeatable in vitro regeneration protocol of puruprea. “The present investigation demonstrates a rapid and repeatable in vitro regeneration protocol of Digitalis purpurea for large scale mass propagation of plantlets”
Ans: In this present study, authors’ primary focus was on to investigate the ploidy status, nuclear DNA content and start codon targeted (SCoT) genetic homogeneity in regenerated Digitalis purpurea (not micropropagation as such). Please see the title of paper. This is the reason micropropagation related discussion (reviewer#3 mentioned in report) is not elaborated and avoided in manuscript. Many useful information has however, been automatically appeared as some of these steps are unavoidable (if not all). Here, in this article, the seed germination process, callus formation from explants, shoot induction and root regeneration steps have been discussed. Our primary job is little different, still we covered and provided photoplates of micropropagation stages (starting from seed germination to outdoor plant transfer through shoot and root induction).The photographic evidence of plants transfer to outdoor (which was omitted earlier) is now added as per reviewer suggestion.
Please, see their conclusions (lines 466-467) "The present investigation demonstrates an efficient in vitro regeneration protocol of Digitalis purpurea for mass propagation of plantlets." This is your first conclusion, you cannot ignore the whole process and the reader believes, with the evidence presented, that an efficient regeneration protocol was established.
Author Response
Please see the attached word file named 'cover letter 2'.

Round 3
Reviewer 3 Report
I still believe that there is no irrefutable evidence that the morphogenetic answer obtained was SE and the answers given to my questions are not entirely convincing. Still there is no irrefutable evidence that the morphogenetic response obtained was SE and the answers to my questions are not convincing.